# What supports and constrains the implementation of multifactorial falls risk assessment and tailored multifactorial falls prevention interventions in acute hospitals? Protocol for a realist review

Rebecca Randell [1,2] Judy M Wright [3] Natasha Alvarado,[1,2] Frances Healey,[4] Dawn Dowding,[5] Heather Smith [6] Nick Hardiker [7] Peter Gardner [2,8] Sue Ward,[9] Chris Todd [5] Hadar Zaman,[8] Lynn McVey,[1,2] Christopher James Davey,[8] David Woodcock[1]

For numbered affiliations see end of article.

Correspondence to
Professor Rebecca Randell;
r.randell@bradford.ac.uk

## ABSTRACT

**Introduction** Falls are the most common type of safety incident reported by acute hospitals and can cause both physical (eg, hip fractures) and non-physical harm (eg, reduced confidence) to patients. It is recommended that, in order to prevent falls in hospital, patients should receive a multifactorial falls risk assessment and be provided with a multifactorial intervention, tailored to address the patient's identified individual risk factors. It is estimated that such an approach could reduce the incidence of inpatient falls by 25%–30% and reduce the annual cost of falls by up to 25%. However, there is substantial unexplained variation between hospitals in the number and type of assessments undertaken and interventions implemented.

**Methods and analysis** A realist review will be undertaken to construct and test programme theories regarding (1) what supports and constrains the implementation of multifactorial falls risk assessment and tailored multifactorial falls prevention interventions in acute hospitals; and (2) how, why, in what contexts and for whom tailored multifactorial falls prevention interventions lead to a reduction in patients' falls risk. We will first identify stakeholders' theories concerning these two topics, searching Medline (1946–present) and Medline In-Process & Other Non-Indexed Citations, Health Management Information Consortium (1983–present) and CINAHL (1981–present). We will then test these theories systematically, using primary studies to determine whether empirical evidence supports, refutes or suggests a revision or addition to the identified theories.

**Ethics and dissemination** The study does not require ethical approval. The review will provide evidence for how to implement multifactorial falls risk assessment and prevention strategies in acute hospital settings. This will be disseminated to academic and clinical audiences and will provide the basis for a future multi-site study through which the theories will be further refined.

**Systematic review registration** PROSPERO CRD42020184458.

### Strengths and limitations of this study

► The use of realist review will allow us to go beyond the question of whether tailored multifactorial falls prevention interventions lead to a reduction in patients' falls risk, to answer questions of how, why, in what contexts and for whom.

► By integrating literature from other settings and concerning interventions with the same mechanisms and using citation searching to identify clusters of related studies, we will ensure we have adequate evidence to provide confidence in our findings.

► We will consult our lay researchers and Study Steering Committee to prioritise the context–mechanism–outcome (CMO) configurations for testing to mitigate against the possibility of the review becoming unwieldy.

► Drawing on a broader range of literature will increase the time required for testing each CMO configuration and may mean that we do not have time to test all the CMO configurations identified.

## INTRODUCTION

Inpatient falls in acute hospitals are an international patient safety concern. Approximately 30%–40% of reported safety incidents in acute hospitals are falls,[1] and in England falls are the most common type of safety incident reported in acute hospitals.[2] Injuries occur in 15%–50% of hospital falls and up to 10% of these are serious.[1] The proportion of falls resulting in any fracture ranges from 1% to 3%, with reports of hip fracture ranging from 1.1% to 2.0%.[3] Outcomes for patients who acquire hip fractures in hospital are far worse than for those who acquire them in

the community, with significant differences in mortality, discharge to long-term high-level nursing care facilities and return to preadmission activity of daily living status.[4]

The human cost of falling also includes fear of falling again and associated loss of confidence,[1 5] loss of independence[6] and social isolation.[2] It can result in slower recovery,[5] even when physical harm is minimal, and can have longer-term consequences for the patient's health, as fear of falling may lead to restriction of activity and associated loss of muscle and balance function, thereby increasing further the risk of falling.[1] Falls can also be a cause of significant distress for families and staff.[3 5] Falls in hospital are a common cause of complaints[7] and can be a source of litigation.[8] Falls in hospital are also associated with increased length of stay and greater amounts of health resource use.[3]

The traditional approach to managing falls in acute hospitals was to complete a falls risk prediction tool (such as St Thomas's risk assessment tool in falling elderly inpatients (STRATIFY)[9]). Such tools typically stratify patients according to their perceived risk of falling (high, medium, low) with interventions targeting individuals at high risk. There are, however, issues with this approach to risk evaluation for falls, in particular the issue of discrimination, where all patients on the unit are identified as high risk, and that having a score provides reassurance that action is being taken when actually it is not.[10]

Given the limitations of risk prediction tools, in the UK, the National Institute for Health and Care Excellence (NICE) guideline on falls in older people states that falls risk prediction tools should not be used and instead a multifactorial falls risk assessment should be undertaken.[11] Rather than categorising a patient according to their perceived risk of falling, this approach to assessment identifies individual risk factors for each patient which may make them at risk of falling and that can be treated, improved or managed during their stay. This may include: cognitive impairment; continence problems; falls history, including causes and consequences (eg, injury and fear of falling); footwear that is unsuitable or missing; health problems that may increase their risk of falling; medication; postural instability, mobility problems and/or balance problems; syncope syndrome; and visual impairment. The NICE guideline states that a multifactorial falls risk assessment should be undertaken for all inpatients 65 years or older and inpatients aged 50–64 years judged to be at higher risk of falling due to an underlying condition. On the basis of this assessment, a multifactorial intervention should be provided, tailored to address the patient's identified individual risk factors. It is estimated that such an approach could reduce the incidence of inpatient falls by 25%–30% and reduce the significant annual cost of falls—estimated at £630 million—by up to 25%.[2]

Even though the NICE guideline has included these recommendations since 2013, there is substantial unexplained variation between National Health Service hospitals in England and Wales, in terms of the number and type of assessments and interventions undertaken.[5] In

assessment, 32% of healthcare providers are still using risk screening tools to identify those at risk of falls.[12] Improvement was found between 2015 and 2017 in the proportion of older patients receiving these assessment and interventions but for some of these remained concerningly low. These include cognitive impairment assessment (58.5%), delirium assessment (39.7%), medications assessment (47.8%) and vision assessment (46.2%). In interventions, there was improvement in the presence of interventions where required for cognitive impairment and delirium, although rates remained low (43.7% and 48.7%, respectively), but no overall significant change in the presence of tailored continence care plans (66.9%) or mobility interventions (78.8%) for those patients who required them.

Given this variation, there is a need to understand the contextual factors that support and constrain the implementation of multifactorial falls risk assessment and tailored multifactorial falls prevention interventions in an acute hospital setting, in order to improve practice. However, even if tailored multifactorial falls prevention interventions are implemented, contextual factors may constrain their use, so that they do not achieve the desired impact. For example, several studies suggest patient adherence to inpatient falls prevention strategies depends on a range of contextual factors including patient willingness to ask for assistance, with some patients not wishing to 'bother' staff[13] or not accepting that they are at risk of falling.[14–17] Therefore, in this paper, we present the protocol for a realist review that aims to determine:

1. What supports and constrains the implementation of multifactorial falls risk assessment and tailored multifactorial falls prevention interventions in acute hospitals; and
2. How, why, in what contexts and for whom tailored multifactorial falls prevention interventions lead to a reduction in patients' falls risk.

This protocol has been written in accordance with Preferred Reporting Items for Systematic Review and Meta-Analysis Protocols (PRISMA-P) guidelines (online supplemental file 1).

## METHODS AND ANALYSIS

We will undertake a realist review. Realist review is a literature review method that represents a divergence from traditional systematic review methodology.[18] It starts by identifying stakeholders' theories and then uses empirical evidence to systematically evaluate these, allowing us to compare how an intervention is intended to work with how it actually works in practice. For realists, interventions do not produce outcomes. Rather, interventions offer resources; outcomes depend on how recipients respond to those resources, which will vary according to the context. Realist theories, referred to as context–mechanism–outcome (CMO) configurations, explain how different contexts trigger particular intervention mechanisms (the reasoning and responses of recipients

to intervention resources) which, in turn, give rise to a particular pattern of outcomes.

Realist approaches can be thought of as consisting of three phases: theory elicitation, theory testing and theory refinement, and we use this structure to describe the process of the realist review.

## Patient and public involvement

The lay member of the project team (DW) contributed to the design of the study. He has recruited a group of lay researchers from different background (members of the public who will contribute to the conduct of this research) who will provide input into the review, prioritising the theories to be tested in phase II of the review. These people (and the lay member of the Study Steering Committee) will draw on their own lived experiences of falling or of caring for someone who has fallen, as well as other life experiences, to ensure that the theories reflect concerns of most importance for patients and carers.

## Phase I: theory elicitation
### Search strategy
Searches will be designed by an information specialist with expertise in realist reviews (JMW) and peer reviewed by a second information specialist. A combination of free-text terms, synonyms and indexing terms will be used. The searches will not be limited by publication date.

The databases to be searched include:
► Ovid Medline (1946–present) and Medline In-Process & Other Non-Indexed Citations
► Ovid Health Management Information Consortium (1983–present)
► EBSCO CINAHL (1981–present)

We will undertake the following searches:
► *Practitioner theories:* programme theories are likely to be found in editorials, comments, letters and news articles,[19] so searches will be undertaken, using a filter (set of search terms) to limit the search to these publication types (see online supplemental file 2 for an example search strategy). In addition to searching the databases listed above, we will search relevant professional journals and the websites of professional organisations. Given the range of professional groups potentially involved in falls risk assessment and prevention, a set of professional journals will be selected covering all the relevant professional groups. This is likely to include, for example, the Nursing Standard, the Pharmaceutical Journal, Frontline (a professional journal for physiotherapists) and Optometry Today. Websites for professional organisations, including the British Geriatrics Society, the Royal College of Nursing, and the Royal Pharmaceutical Society, will be searched. Searches will also be run on Google for reports of quality improvement projects, such as the FallSafe quality improvement project.[7]
► *Academic theories:* the discussion sections of empirical studies often include the authors' theories about why the intervention did or did not achieve the desired effect.[20] Therefore, studies of falls prevention interventions will be searched for, using existing systematic reviews as a starting point. See online supplemental file 3 for an example search strategy.
► *Substantive theories:* we will review articles retrieved in the 'academic theories' search for reference to substantive theory and, if necessary, we will undertake an additional search for relevant substantive theories on risk assessment and guideline adherence.

The records identified in the searches will be saved and managed in an EndNote library. Details of all search activities (databases, websites, date of search, number of records found, search strategies) will be recorded in a timeline spreadsheet.

### Screening process and inclusion/exclusion criteria
A 'liberal accelerated' approach to screening will be taken, where one reviewer reviews all records/full-text papers and a second reviewer reviews records/full-text papers excluded by the first reviewer.[21] This approach is less time and resource intensive than having two reviewers review all records/full-text papers while maximising inclusion, increasing the number of records/full-text papers retained in comparison to a single reviewer.[22] Because the purpose of this phase of the review is to identify and catalogue programme theories and theory fragments, rather than to assess their validity, selection will be based on relevance to the topic of the review.[18 19] The inclusion criteria for the 'practitioner theories' and 'academic theories' searches will be:
► Multifactorial/single-factor falls risk assessment or falls risk prediction tools and/or multifactorial/single falls prevention interventions
► Adults/older people.
► Acute hospital setting
► Include arguments about what supports or constrains implementation and/or in what contexts and for whom they can/should be used.
► Published in the English language.

Exclusion criteria will be:
► Children and young people
► Settings other than acute hospital
► Published in languages other than English.

We will include articles about single-factor risk assessment tools on the basis that understanding what supports and constrains their use will inform our understanding of what supports and constrains use of multifactorial risk assessment tools. Similarly, we will include articles about single falls prevention interventions on the basis that understanding what supports and constrains the implementation and use of single interventions will inform our understanding of what supports and constrains the implementation and use of multifactorial interventions which contain those single interventions as a component. This is in line with the realist approach, which seeks to link the responses to an intervention to particular resources provided by the intervention. We will also include articles about falls risk prediction tools to understand how and in

what contexts they continue to be used instead of multifactorial falls risk assessments.

We will exclude articles published in languages other than English because the nature of realist review means that we would need to translate the full article, for which the project does not have adequate resources. This is in contrast to traditional systematic reviews where only defined data needs to be identified and translated.[23]

A PRISMA flow chart detailing the review decision process for phase I will be developed.

### Analysis and synthesis

Included articles from the 'practitioner theories' and 'academic theories' searches will be imported into NVivo and coded as context, mechanism and outcome. Outcomes will include, for example, fall rates, but also any other outcomes reported, to capture both intended and unintended impacts. A 10% random sample of papers will be coded by a second reviewer for consistency. A Microsoft 365 Excel spreadsheet will be used for recording the CMO configurations from each article. Our experience of undertaking realist reviews suggests that individual articles are unlikely to provide us with fully formed CMO configurations or to even contain information about all three elements of context, mechanism and outcome.[20] Therefore, alongside recording any complete CMO configurations that we identify, we will also record CMO fragments in the Excel spreadsheet. Once this is complete, the list of CMO configurations will be refined to combine those that are similar. Narrative summaries of each of the substantive theories identified will be written and we will compare the CMO configurations with the substantive theories, using the substantive theories to fill in any remaining gaps in the CMO configurations. The resulting CMO configurations, explaining both[1] what supports and constrains implementation of multifactorial falls risk assessment and tailored multifactorial falls prevention interventions in acute hospitals, and[2] how, why, in what contexts, and for whom tailored multifactorial falls prevention interventions lead to a reduction in patients' falls risk, will combine to provide an initial programme theory.

A particular risk for realist reviews is that they can easily become unwieldy.[19] We will mitigate against this by taking guidance from our lay researchers and Study Steering Committee regarding the CMO configurations that should be taken forward for testing in phase II of the review. We will first identify a subset of possible CMO configurations, based on the feasibility of testing them, undertaking initial scoping searches to gauge the extent of the available literature, and based on their potential for informing practice (eg, if a CMO configuration contains contextual factors that constrain the conduct of falls risk assessment that are not amenable to change, it will not be taken forward for testing). We will discuss the remaining subset of CMO configurations with our lay researchers and our Study Steering Committee, which brings together clinicians and academics with expertise including falls prevention, risk assessment, patient safety and implementation science. We will ask them to rank the CMO configurations in order of priority; those which have the highest ranking across both groups will be taken forward to the next stage.

### Phase II: theory testing
#### Search strategy
Searching will be purposive and iterative, driven by the prioritised CMO configurations, in order to identify empirical studies relevant to testing of the initial programme theory.[19] Searches will be designed by an information specialist (JMW) with input from the review team. It will be peer reviewed by a second information specialist. Health and multidisciplinary databases to be searched include:

- ▶ Ovid Medline and Medline In-Process & Other Non-Indexed Citations (1946–present)
- ▶ EBSCO CINAHL (1981–present)
- ▶ Ovid EMBASE (1947–present)
- ▶ Web of Science Core Collection (1900–present)
- ▶ ProQuest Applied Social Sciences Index & Abstracts (1987–present)

An initial scoping search suggests there is limited empirical evidence from the hospital setting, with existing research tending to focus on the community setting. However, realist reviews offer particular benefits when considering interventions where there is limited primary research because the key unit of analysis is the intervention mechanism; this means that literature concerning the same intervention in another setting or other interventions that have the same underlying mechanism are deemed relevant, so a wider breadth of evidence is available.[19 24] Consequently, while initial searches will be limited to the hospital setting, where there is an absence of literature searches will be broadened out to include literature from the community setting and care homes. We may also broaden our search to include literature concerning other interventions that are based on the same mechanisms as those within the initial programme theory. Search techniques will include structured literature searching of academic databases listed, and also complementary searching such as citation searching and other CLUSTER (Citations, Lead authors, Unpublished materials, Scholar searches, Theories, Early examples, and Related projects) searching techniques[25] that can identify relevant studies through links in citation networks or through a focus on specific authors or projects. Grey literature searching (for example websites of professional organisations) will be undertaken where it likely to uncover literature relevant to the programme theories under investigation.

#### Screening process
As in phase I, a 'liberal accelerated' approach to screening will be taken. Relevance of each study to testing the initial programme theory will be assessed pragmatically against key inclusion criteria concerned with the context (acute

hospitals) and the intervention (falls risk assessment and/or falls prevention interventions). Priority will be given to those studies that meet all inclusion criteria but we will also include studies which match the intervention criteria but not the context criteria (eg, studies about falls risk assessment in care homes) and studies which match the context criteria and are concerned with interventions that have the same underlying mechanism (eg, studies about pressure ulcer risk assessment in acute hospitals). All study designs will be included, acknowledging that different study designs make different contributions to theory testing; for example, randomised controlled trials provide information on outcome patterns and may provide some pointers to likely contextual differences, but they seldom provide information about mechanisms, information which is more likely to be found in qualitative studies. A PRISMA flow chart detailing the review decision process for phase II will be developed.

### Appraisal and analysis

Studies deemed to be relevant will be appraised using the Mixed-Methods Appraisal Tool.[26] However, we will not exclude studies based on this appraisal. Additionally, following the realist approach, in describing the studies, we will reflect only on the quality of those elements of the studies from which *evidential fragments* for theory testing are drawn.[27] For example, in a mixed-methods study, questionable analyses of falls data are not of concern if what we are drawing on are the well-conducted qualitative elements of the study. As in phase I, included studies will be imported into NVivo and coded as context, mechanism, and outcome, capturing all reported outcomes.

Guidelines for systematic reviews suggest that, in addition to assessing risk of bias in individual studies, an assessment of the risk of bias across studies—such as publication bias and selective reporting within studies—should be undertaken.[28] However, this assumes a traditional systematic review that relies on quantitative studies and uses approaches that are not easily applicable when using the wide range of study designs that realist reviews typically incorporate.

### Phase III: theory refinement

Coded data for each individual study will be compared in turn with the initial programme theory to determine whether the findings support, refute or suggest a revision or addition to the CMO configurations. The resulting programme theory will be summarised in both diagrammatic and narrative form.[29 30]

In reporting the review, the Realist And Meta-narrative Evidence Syntheses: Evolving Standards publication standards will be followed.[31] Any changes from this protocol will be reported and the rationale provided.

For systematic reviews, it is recommended that the strength of the body of evidence is assessed and reported,[28] for example using the Grades of Recommendation, Assessment, Development and Evaluation (GRADE) approach[32] or GRADE-CERQual (Confidence in the Evidence from Reviews of Qualitative research).[33] Approaches such as GRADE are not appropriate for a realist review, because they rely on hierarchies of evidence in making assessments and treat inconsistency in effects across studies as a problem, whereas realist reviews accept that there may be 'nuggets of wisdom' in methodologically weak studies[27] and expect variation in effects because of variation in programme contexts.[34] GRADE-CERQual involves assessing each individual review finding based on the four components of methodological limitations, coherence, adequacy of data and relevance.[33] It has been used for previous realist reviews[35 36] and fits better with the realist approach, involving consideration of the theoretical contributions of studies and encouraging reviewers to be sensitive to the importance of context.[14] Therefore, we will use CERQual to assess each CMO, rating confidence in each as either high, moderate, low or very low. This will both support decision-making of those who wish to use the findings of the review to inform their practice and highlight areas where further primary research is needed.

## ETHICS AND DISSEMINATION

Ethical approval is not required for this review.

This review will provide evidence that healthcare providers can use to inform their own multifactorial falls risk assessment and prevention strategies, with the potential to reduce frequency of inpatient falls and thereby reduce the impact of both human suffering and healthcare costs. Therefore, the results will be published in an academic journal that has a clinical readership. We will also present the findings at other venues where we will reach clinical staff, including the Royal College of Nursing International Nursing Research Conference, local Falls Collaboratives, and Nursing, Midwifery and AHP Research conferences at local trusts. We will engage with the wider public via a project website, where links to publications will be provided, and social media, for example, Twitter.

**Author affiliations**
[1]Faculty of Health Studies, University of Bradford, Bradford, UK
[2]Wolfson Centre for Applied Health Research, Bradford, UK
[3]Institute of Health Sciences, University of Leeds, Leeds, UK
[4]NHS England and NHS Improvement, London, UK
[5]Division of Nursing, Midwifery & Social Work, The University of Manchester, Manchester, UK
[6]Medicines Management & Pharmacy Services, Leeds Teaching Hospitals NHS Trust, Leeds, UK
[7]School of Human and Health Sciences, University of Huddersfield, Huddersfield, UK
[8]Faculty of Life Sciences, University of Bradford, Bradford, UK
[9]Manchester University NHS Foundation Trust, Manchester, UK

**Contributors** RR is principal investigator and guarantor for the review. She conceived, designed and secured funding for the review in collaboration with JMW, NA, FH, DD, HS, NH, PG, SW, CT and DW. All authors provided input into various aspects of the design of the review. In particular, JMW developed search strategies and advised on methods for assessing strength of evidence. NA advised on realist methods. FH and CT advised on literature on falls. HS and SW advised on current

practices for falls risk assessment in the NHS. FH, DD, HS, NH, PG, CT, HZ and CJD advised on appropriate journals to search. DW, RR, NA and LM developed the plan for lay researcher involvement in the study, and all authors contributed to clarifying and refining inclusion and exclusion criteria. RR led the writing of this protocol manuscript and all authors have revised drafts and read and approved the final manuscript.

**Funding** This research is funded by the National Institute for Health Research (NIHR) Health Services and Delivery Research (HS&DR) Programme (project number NIHR129488).

**Disclaimer** The views and opinions expressed are those of the author and do not necessarily reflect those of the HS&DR Programme, NIHR, NHS or the Department of Health.

**Competing interests** None declared.

**Patient consent for publication** Not required.

**Provenance and peer review** Not commissioned; externally peer reviewed.

**ORCID iDs**
Rebecca Randell http://orcid.org/0000-0002-5856-4912
Judy M Wright http://orcid.org/0000-0002-5239-0173
Heather Smith http://orcid.org/0000-0001-9740-4594
Nick Hardiker http://orcid.org/0000-0002-7629-5664
Peter Gardner http://orcid.org/0000-0002-8799-0443
Chris Todd http://orcid.org/0000-0001-6645-4505

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
