## [Reviewer comments · BMJ Open]

ARTICLE DETAILS

TITLE (PROVISIONAL)	What supports and constrains the implementation of multifactorial falls risk assessment and tailored multifactorial falls prevention interventions in acute hospitals? Protocol for a realist review
AUTHORS	Randell, Rebecca; Wright, Judy; Alvarado, Natasha; Healey, Frances; Dowding, Dawn; Smith, Heather; Hardiker, Nick; Gardner, Peter; Ward, Sue; Todd, Chris; Zaman, Hadar; McVey, Lynn; Davey, Christopher; Woodcock, David

VERSION 1 – REVIEW

REVIEWER	Martin, Rachelle Burwood Academy of Independent Living
REVIEW RETURNED	03-May-2021

GENERAL COMMENTS	Thank you for the opportunity to review this protocol. Realist review methods will provide helpful information to better understand what supports and constrains the implementation of multifactorial falls risk assessment and falls prevention interventions in acute hospitals. The protocol aligns excellently with realist review methods and guidance standards. I would like to understand more about the lived experience and/or expertise of the lay members and Study Steering Committee. Given that these groups will be playing a pivotal role in focusing the scope of the review and prioritising CMO configurations, more information about their ability to support this decision-making process would be helpful. In the analysis section, you state you will record CMO fragments and then later combine these. However, at times, CMO linkages can be found in the source data. It would be helpful to better understand how you will ensure that these important linkages are maintained within the coding process. Thank you again for the opportunity to review. I look forward to reading your findings in the near future.
--

VERSION 1 – AUTHOR RESPONSE

Reviewer 1's comments:

- I would like to understand more about the lived experience and/or expertise of the lay members and Study Steering Committee. Given that these groups will be playing a pivotal role in focusing the scope of the review and prioritising CMO configurations, more information about their ability to support this decision-making process would be helpful.

In the 'Patient and public involvement' section, we now give more information about the experience of our lay researchers. At the end of the description of Phase 1, we have provided further details of our Study Steering Committee, including their areas of expertise.

- In the analysis section, you state you will record CMO fragments and then later combine these. However, at times, CMO linkages can be found in the source data. It would be helpful to better understand how you will ensure that these important linkages are maintained within the coding process.

We have now revised the wording to make clear that, where identified, we will record any complete CMO configurations.

VERSION 2 – REVIEW

REVIEWER	Martin, Rachelle Burwood Academy of Independent Living
REVIEW RETURNED	06-Jul-2021
GENERAL COMMENTS	Thank you for the opportunity to consider the author's responses to peer review comments. They have addressed the issues raised, and I am very happy to recommend that this version of the manuscript be accepted for publication.